# Nocturnal Hypoxemia and CT Determined Pulmonary Artery Enlargement in Smokers

**DOI:** 10.3390/jcm10030489

**Published:** 2021-01-30

**Authors:** Marta Marin-Oto, Luis M. Seijo, Miguel Divo, Gorka Bastarrika, Ana Ezponda, Marta Calvo, Javier J. Zulueta, Guillermo Gallardo, Elena Cabezas, German Peces-Barba, Maria T. Pérez-Warnisher, Jose M. Marín, Bartolomé R. Celli, Ciro Casanova, Juan P. De-Torres

**Affiliations:** 1Department of Respiratory Medicine, Clínica Universidad de Navarra, University of Navarra, Avenida Pío XII, 36, 31008 Pamplona, Spain; jzulueta@unav.es; 2Department of Respiratory Medicine, Clínica Universidad de Navarra, University of Navarra, Calle Marquesado de Sta. Marta, 1, 28027 Madrid, Spain; lseijo@unav.es (L.M.S.); mperezw@unav.es (M.T.P.-W.); 3Pulmonary and Critical Care Division, Brigham and Women’s Hospital, Harvard Medical School, 75 Francis St., Boston, MA 02115, USA; MDivo@copdnet.org (M.D.); BCelli@copdnet.org (B.R.C.); 4Department of Radiology, Clínica Universidad de Navarra, University of Navarra, Avenida Pío XII, 36, 31008 Pamplona, Spain; bastarrika@unav.es (G.B.); aezponda@unav.es (A.E.); mcalvoi@unav.es (M.C.); 5Department of Respiratory Medicine, Fundación Jiménez-Díaz, Av. de los Reyes Católicos, 2, 28040 Madrid, Spain; ggallardom@unav.es (G.G.); elena.cabezas@quironsalud.es (E.C.); gpeces@fjd.es (G.P.-B.); 6Department of Respiratory Medicine, Hospital Universitario Miguel Servet, University of Zaragoza, Av. Isabel la Católica 3, 50009 Zaragoza, Spain; jmmarint@unizar.es; 7Department of Respiratory Medicine, Hospital Nuestra Señora de la Candelaria, Ctra. Gral. del Rosario, 145, 38010 Santa Cruz de Tenerife, Spain; casanovaciro@gmail.com; 8Pulmonary Division, Kingston General Hospital, Queen’s University, 76 Stuart St., Kingston, ON K7L 2V7, Canada; jupa65@hotmail.com

**Keywords:** pulmonary artery enlargement, COPD, sleep-disordered breathing, nocturnal hypoxemia, chest CT scan, oximetry, personalised medicine, clinical phenotypes

## Abstract

Background: Pulmonary artery enlargement (PAE) detected using chest computed tomography (CT) is associated with poor outcomes in chronic obstructive pulmonary disease (COPD). It is unknown whether nocturnal hypoxemia occurring in smokers, with or without COPD, obstructive sleep apnoea (OSA) or their overlap, may be associated with PAE assessed by chest CT. Methods: We analysed data from two prospective cohort studies that enrolled 284 smokers in lung cancer screening programs and completing baseline home sleep studies and chest CT scans. Main pulmonary artery diameter (PAD) and the ratio of the PAD to that of the aorta (PA:Ao ratio) were measured. PAE was defined as a PAD ≥ 29 mm in men and ≥27 mm in women or as a PA:Ao ratio > 0.9. We evaluated the association of PAE with baseline characteristics using multivariate logistic models. Results: PAE prevalence was 27% as defined by PAD measurements and 11.6% by the PA:Ao ratio. A body mass index ≥ 30 kg/m^2^ (OR 2.01; 95%CI 1.06–3.78), lower % predicted of forced expiratory volume in one second (FEV_1_) (OR 1.03; 95%CI 1.02–1.05) and higher % of sleep time with O_2_ saturation < 90% (T90) (OR 1.02; 95%CI 1.00–1.03), were associated with PAE as determined by PAD. However, only T90 remained significantly associated with PAE as defined by the PA:Ao ratio (OR 1.02; 95%CI 1.01–1.03). In the subset group without OSA, only T90 remains associated with PAE, whether defined by PAD measurement (OR 1.02; 95%CI 1.01–1.03) or PA:Ao ratio (OR 1.04; 95%CI 1.01–1.07). Conclusions: In smokers with or without COPD, nocturnal hypoxemia was associated with PAE independently of OSA coexistence.

## 1. Introduction

Chronic obstructive pulmonary disease (COPD) and sleep-disordered breathing (SDB) are associated with extensive morbidity and mortality, mainly due to a worldwide rise in the prevalence of smoking, obesity and aging [1,2]. The coexistence of obstructive sleep apnoea (OSA) and COPD is defined as the overlap syndrome (OVS) [3], and is associated in cohort studies with poorer outcomes [4].

During sleep, COPD, OSA and their overlap may present with varying degrees of nocturnal hypoxemia (NH). The coexistence of COPD and OSA may have a synergistic adverse effect on pulmonary haemodynamics leading to right ventricular dysfunction and pulmonary hypertension (PH) [5]. COPD is a well-known cause of PH and cor pulmonale with chronic hypoxia as the principal underlying mechanism. Home oxygen therapy in patients with COPD is often prescribed for nocturnal use and in selected patients, it can reduce mean pulmonary artery pressures (mPAP) [6] and mortality [7,8]. In severe OSA without concomitant cardiopulmonary disease, pulmonary artery pressures may be elevated but are generally mild [9,10,11]. Because invasive right heart catheterisation (RHC) is the gold standard for the diagnosis of PH, there is a dearth of studies investigating the prevalence of PH in COPD, OSA and OVS.

Chest computed tomography (CT) has emerged as a non-invasive tool capable of identifying PH. Pulmonary artery diameter (PAD), alone or in combination with aortic diameter (PA:Ao ratio), can be measured to determine the presence of pulmonary artery enlargement (PAE). These measurements correlate with mPAP values obtained during RHC [12], and are clinically relevant. It is has been described that the presence of PAE in COPD patients is associated with increased risk of exacerbation [13] and worse survival [14]. We hypothesised that in smokers with or without COPD, the presence of NH would be associated with PAE independently of the coexistence of OSA.

## 2. Methods

### 2.1. Study Population

We included in this analysis, all participants recruited from the Sleep Apnoea in Lung Cancer Screening Study (SAILS) cohort (Madrid, Spain) [15] and from the COPD History Assessment in Spain (CHAIN) cohort (Pamplona site, Spain) [16] between December 2015 and December 2019. Details of the inclusion and exclusion criteria for both cohorts have been described elsewhere [15,16]. Briefly, the SAILS study was a prospective study that evaluated the prevalence of OSA in a lung cancer screening program (ClinicalTrials.gov Identifier: NCT02764866), whereas the CHAIN study is a multicentre prospective study that recruited smokers with/without COPD to better define its natural history and potential clinical phenotypes (ClinicalTrials.gov Identifier: NCT01122758). Additional details on inclusion/exclusion criteria are reported in the Appendix A. The research protocols were approved by the ethics committees of the participating canters (IRB# 23/2014-FJD and IRB # 28/2012, respectively). All participants gave their written informed consent.

### 2.2. Measurements

Demographic and anthropometric measurements were obtained in all patients during the initial visit. Pulmonary function tests, a home sleep study (HSS) and chest CT imaging were subsequently performed. Spirometry and lung diffusion capacity for carbon monoxide (DLCO) testing complied with ERS/ATS guidelines [17,18]. In both cohorts, all patients had a cumulative smoking exposure >10 packs/year and the diagnosis and staging of COPD were established according to Global Initiative for Chronic Obstructive Lung Disease recommendations [19].

### 2.3. Home Sleep Study

All participants completed the Epworth Sleepiness Scale (ESS) questionnaire [20]. A single night HSS was performed with a portable sleep monitor. All studies were manually analysed in a blinded fashion according to the American Academy of Sleep Medicine standards [21]. Recorded data included nasal-oral airflow, chest and abdominal wall motion, snoring and arterial oxygen saturation by pulse oximetry (SpO_2_). Apnoea was defined as an at least 90% decrease in the oronasal pressure signal and hypopnoea was defined as an at least 30% decrease in the oronasal pressure signal combined with ≥ 3% arterial oxygen desaturation lasting at least 10 s. Time elapsed with SpO_2_ below 90% was recorded and expressed as a percentage time with O_2_Sat below 90% over total valid recorded time (T90). The oxygen desaturation index (ODI ≥ 3%) and apnoea–hypopnoea index (AHI) were calculated.

### 2.4. Chest CT

Baseline chest CT scans were interpreted in a blinded fashion by 2 radiologists (G.B., A.E.). Details of CT image acquisition and reconstruction protocols have been described elsewhere [14]. The diameter of the main PAD at the level of its bifurcation and the diameter of the ascending aorta (Ao) in its maximum dimension were measured using the same images (Appendix A). To define PAE, we used The Framingham Heart Study normative values for PAD by CT that established sex-specific cut-off values for men of 29 mm (26 ± 2.9 mm) and 27 mm for women (24.2 ± 2.7 mm) as well as the normative value for PA:Ao ratio of 0.9 for both sexes [22].

### 2.5. Statistical Analysis

Data were summarised as mean and standard deviation (SD) or median and interquartile range (IQR) for continuous variables according to their distribution. Categorical or binomial variables are presented as proportions. We used two-sample *t*-tests, Pearson’s chi-squared tests to compare between two groups, depending on the type of variable and their distribution. Unadjusted and multivariable logistic regression models were used to examine PAE’s predictors. Covariates for the model included demographics (e.g., sex, age, body mass index—BMI—categorised as “non-obese”—BMI < 30 kg/m^2^—vs. “obese”—≥ 30 kg/m^2^, smoking history, study site), clinical variables (prevalent comorbidities, predicted forced expiratory volume in one second (FEV_1_) and DLCO), sleep variables (resting O_2_ saturation, AHI or ODI, T90%) and radiological parameters (% of lung with emphysema). The estimate of association was expressed as adjusted odds ratio (aOR) with corresponding confidence intervals of 95%. To assess the effect of SpO_2_ measures from the sleep recordings (e.g., T90 or median SO_2_), only one of the hypoxemia measures was included in each tested model to avoid interaction. To assess the robustness of the results, we performed several sensitivity analyses for the different cut-offs of significant NH (T90 > 10% and T90 > 30%) and different definitions of PAE (e.g., PA:Ao ratio of >0.9). Analyses were performed using STATA/SE 14.1 (StataCorp, College Station, TX, USA), and a two-sided *p*-value less than 0.05 was considered significant.

## 3. Results

### 3.1. Participant Characteristics

During the recruitment period, 327 subjects met the inclusion criteria, 238 from the SAILS study and 89 from the Pamplona site of the CHAIN study. Of those, 43 were excluded due to resting hypoxemia, presence of aortic aneurism and/or interstitial lung disease or inaccurate chest CT scans or HSS studies (Figure 1). The characteristics of the 284 participants included in the analysis are shown in Table 1. The mean (±SD) age of participants was 65 ± 7 years and 63% of them were male. There were 167 (59%) individuals with OSA as defined by an AHI > 5 events·h^−1^, and 171 (60%) with COPD. The characteristics of the participants from each site are shown in Appendix A. Sex distribution and prevalent comorbidities including OSA were similar among the two cohorts. The CHAIN cohort was slightly younger (63 vs. 66 years of age) and included a higher proportion of participants with COPD (66% vs. 52%) (Appendix A).

### 3.2. Prevalence and Characteristics of Patients with Pulmonary Artery Enlargement

The distribution of the observed PAD is shown in Figure 2. Median PAD values in women and men were 25 mm (interquartile range—IQR = 23 to 26 mm) and 26 mm (IQR = 24 to 30 mm), respectively. Twenty-six women (24.5%) and 49 men (27.5%) met the predetermined criteria for PAE. The prevalence rates of PAE for smokers without OSA or COPD, with OSA alone, with COPD alone or with OVS were 26, 24, 28 and 28%, respectively. Table 1 describes the characteristics of subjects with and without PAE as defined by PAD measurements. Participants with or without PAE did not differ in age, sex distribution or comorbidities. The group with PAE had a higher tobacco consumption (mean 50 vs. 40 packs/year, *p* = 0.027), higher BMI (median 30 kg/m^2^ vs. 27 kg/m^2^, *p* = 0.002), lower resting O_2_ saturation (median 93% vs. 94%, *p* = 0.002) and poorer lung function as assessed by the post-bronchodilator % predicted FEV_1_ (mean 74% vs. 88% predicted, *p* < 0.001). No difference was found in the prevalence of COPD (63% vs. 59%, *p* = 0.613) or OSA (57% vs. 59%, *p* = 0.763).

Seventeen men (9.5%) and 15 women (14.1%) had a PA:Ao ratio > 0.9 (Figure 2). The prevalence rates of PAE based on PA:Ao ratio for smokers without OSA or COPD, with OSA alone, with COPD alone or with OVS were 11, 10, 11 and 12%, respectively. The characteristics of participants grouped by the presence of PAE as assessed by the PA:Ao ratio are shown in Appendix A. Subjects with PAE according to the PA:Ao ratio criterion, did not differ in sex distribution, age, smoking history, BMI, percentage of lung volume with emphysema on CT and prevalent COPD or OSA. The group with a PA:Ao ratio > 0.9 had lower resting O_2_ saturation (93% vs. 94%, *p* = 0.026), poorer lung function as assessed by FEV_1_ (77% vs. 85% predicted, *p* = 0.031) and more NH as assessed by median SpO_2_ (90% vs. 92%, *p* < 0.001) and T90 (27% vs. 5%, *p* = 0.009) compared with the group with a PA:Ao ratio ≤ 0.9.

The distribution of COPD and OSA patients according with GOLD classification and AHI index, respectively, is shown in Appendix A. As expected, severe COPD had a higher PAE as assessed by PAD prevalence than mild–moderate COPD but there were no differences when PAE was defined by PA:Ao. There were no differences in PAE prevalence among the cases of OSA classified according to AHI index.

### 3.3. Association between Nocturnal Hypoxemia and Pulmonary Artery Enlargement

Pulmonary artery diameter increased with percentage of time spent with an SpO_2_ < 90% in both men and women (Appendix A). We found significant univariate associations between PAE, as defined by PAD measurements, and smoking history, BMI, lower % predicted FEV_1_ and resting O_2_ saturation. Among respiratory variables obtained during the sleep recording, PAE was associated with a higher ODI, median nocturnal SpO_2_ and T90 (Appendix A). Multiple logistic regression showed independent associations between PAE as defined by PAD measurements and obesity (OR 2.01; 95% CI 1.06–3.78; *p* = 0.031), higher T90 (OR 1.02; 95% CI 1.00–1.03; *p* = 0.007) and lower FEV_1_ and median SpO_2_ sleep recording values (OR 1.03; 95% CI 1.02–1.05; *p* = 0.004 and OR 1.16; 95% CI 1.02–1.30; *p* = 0.038) (Figure 3A). Because there is no consensus definition for significant NH as assessed by T90, we also evaluated in separated models two thresholds of T90, T90 > 10% and T90 > 30% for a potential association with PAE. Fully adjusted T90 > 30% was associated with PAE (OR 2.61; 95% CI 1.29–5.04; *p* = 0.008) (Figure 3A). In multiple logistic-regression models only NH values were independently associated with a PA:Ao ratio > 0.9, including lower median SpO_2_ (OR 1.21; 95% CI 1.02–1.41; *p* = 0.036), higher T90 (OR 1.02; 95% CI 1.01–1.03; *p* = 0.004) and T90 ≥ 30% (OR 2.92; 95% CI 1.16–7.53; *p* = 0.023) (Figure 3B).

### 3.4. Sensitivity Analysis

To further assess the association of NH not related with apnoeic events, we excluded from the analysis individuals with OSA. In this subgroup of 117 smokers, including 80 with COPD, 32 subjects had PAE (27%). A reduction in lung function tended to be related with PAE as assessed by PAD (*p* = 0.059), however in adjusted analysis, only T90 or alternatively T90 ≥ 30% were associated with PAE with odds ratios (95% CI) of 1.02 (1.01–1.04) and 3.63 (1.23–9.07), respectively (Appendix A). For its part, PAE as defined by PA:Ao ratio, was only associated with NH measures such as median SpO_2_ (OR 1.38; 95% CI 1.05–1.72), T90 (OR 1.02; 95% CI 1.01–1.07) or T30 (OR 3.75; 95% CI 1.28–9.02) (Appendix A). To test if intermittent NH, as reflected by the ODI, was associated with PAE as defined by PAD measurements or PA:Ao, we repeated the analysis replacing AHI by ODI. After adjustment, ODI failed to perform better than AHI.

## 4. Discussion

In this study of smokers with or without COPD, we found that the presence of NH was associated with a higher prevalence of PAE according to chest CT scan, independent of coexisting OSA. The prevalence of PAE measured by chest CT ranged from 11.3 to 26.4%, depending on the methodology used to define PAE. The latter is independently associated with NH, BMI, and worse lung function if PAD sex-specific criteria are used, and only with NH if the PA:Ao ratio is used to define it. These findings could be relevant for clinical practice since neither chest CT nor sleep studies are currently recommended in the routine management of COPD patients [19].

Several seminal studies have described the development of PH in patients with advanced COPD [23]. These patients are classified within group 3 (PH with lung diseases and/or hypoxia) by the Sixth World Symposium on Pulmonary Hypertension (WSPH); if FEV_1_ is < 60%, predicted or severe structural changes are present in the lung parenchyma as assessed by thin-slice CT [24]. COPD patients with an FEV_1_ > 60% predicted with PH are assigned to group 1 (pulmonary artery hypertension). With the increasing use of non-invasive techniques allowing a reliable estimation of mPAP, many patients with mild to moderate COPD, and smokers without COPD, meet criteria for group 1 of the WSPH [13,25,26]. Our findings confirm previous reports of a high prevalence of PAE in smokers, and are clinically relevant, since PAE determined by imaging techniques is associated with an increase in morbidity and mortality in a number of clinical settings, including patients with mild or moderate COPD [14], smokers without COPD [27] and even patients with pneumonia due to COVID-19 [28].

Unfortunately, the two criteria commonly used to determine the presence of PAE in the Framingham study do not lead to equivalent results. In our study, and based on PAD measurements for men and women, the prevalence rates of PAE for smokers without OSA or COPD, with OSA alone, with COPD alone or with OVS were 26, 24, 28 and 28%, respectively, whereas the prevalence of PAE was much lower in all groups when using the PA:Ao ratio (11, 10, 11 and 12%, respectively). That notwithstanding, PAE in our study was much more common when compared to the 4.2% rate obtained in smokers enrolled in the Mount Sinai Early Lung and Cardiac Action Program [29]. This discrepancy may be due to the cut-off value used to define PAE in that study (PAD ≥ 34 mm). However, the prevalence of PAE, as defined by the PA:Ao ratio in smokers without OSA or COPD, was 8.1%, similar to the 6.9% reported by Steiger et al., that used a cut-off of PA:Ao ≥ 1.0 [29]. Unfortunately, these authors did not report lung function or sleep related data so the potential contribution to PAE related to the coexistence of airflow obstruction or sleep-disordered breathing was not assessed.

In line with other studies, we found that PAE as defined by PAD, after adjusting for age and sex, was also associated with BMI above 30 kg/m^2^ [14,22,29], and lower FEV_1_ [14]. The present study findings also agree with a recent analysis of the 3464 participants in the COPDGene study that showed a mild association between BMI and decreased FEV1 with PAE [13]. However, the COPDGene study did not register resting SaO_2_ saturation nor sleep recordings. This was not the case when PAE was assessed by the PA:Ao ratio. As previously reported in the Framingham study [22], we found that PAE defined by the PA:Ao ratio was independent of obesity or FEV_1_, with nocturnal hypoxemia being the only variable that showed a link with PAE in the full adjusted model.

In the present analysis, we found a robust association between NH and PAE defined by both the PAD and the PA:Ao ratio using models that were adjusted for both lung function and sleep-disordered breathing. In addition, a secondary analysis to test for effect modification of the presence of OSA, showed no evidence that our results were different in smokers with or without OSA. However, our study demonstrates that in smokers, poorer lung function (even mild–moderate) but not OSA, is independently associated with increased PAD, suggesting that isolated or non-intermittent NH probably plays a more prominent role than intermittent hypoxia in the pathogenesis of PH in smokers with COPD and/or OSA. On the other hand, the relationship between NH and PAE seems to be dose dependent. A T90 > 30% cut-off was independently associated with PAE, as defined by either method, in all smokers as well as those without OSA that had mostly isolated or non-intermittent NH.

The co-existence of airflow obstruction as the main determinant of nocturnal hypoxemia in patients with OSA has been emphasised in a recent study [30]. In that study, OSA patients with or without COPD had the same AHI. However, mean SpO_2_ during sleep in subjects with OVS was much lower [30]. This is important because previous studies demonstrated that the severity of NH, but not the AHI, correlated with mPAP RHC measurements in OVS [5]. Our results support the hypothesis that NH, mainly non-intermittent NH, is the putative sleep variable associated with a higher risk of PAE in smokers. Unfortunately, there has been little interest in the evaluation of the level of SaO_2_ during sleep in COPD until recently. In a large retrospective study, Kendzerska et al., demonstrated that the degree of NH had a better ability to predict incident PH than AHI did in individuals with OVS [31]. On the other hand, in a recent report, the presence of NH in a group of COPD patients was associated with higher serum C-reactive protein levels, a well-known risk factor for cardiovascular disease [32]. Together with our results, all of these studies provide evidence that nocturnal oximetry should be evaluated in smokers with COPD and evidence of PAE on chest CT scans. Effective treatment with nocturnal positive pressure ventilation in OVS patients improves outcomes [4]. However, data on the potential benefits of nocturnal oxygen therapy in patients with isolated NH are lacking. In a randomised trial of long-term oxygen therapy for COPD, patients had moderate daytime hypoxemia, but sleep recordings were not performed, therefore it was not possible to determine if non-intermittent NH or OSA were present [33]. In another recent underpowered trial of nocturnal oxygen in COPD with isolated NH, it was concluded that this therapy does not shown a clear benefit [34]. In the later study, sleep apnoea was excluded in most subjects, however CT scan were not performed in any of the two trials, so the role of the PAE on the effect of oxygen therapy was not assessed and it remains unclear which patients stand to benefit the most.

Our results fit with previous findings that demonstrated that smokers without COPD or animals chronically exposed to cigarette smoke (CS) can develop PH [35]. In fact, among smokers without COPD or OSA, 26 and 11% had PAE defined by the diameter of the pulmonary artery or by the PA:Ao ratio, respectively. We can only speculate that there must be a genetically based response to the effect of tobacco smoke on the pulmonary vasculature. It is well known that hypoxia drives pulmonary hypertension (PH) by promoting pulmonary arterial remodelling and dysregulation of voltage-gated K+ channels. Interestingly, Sevilla-Montero et al. [36] showed that cigarette smoking contributes to pulmonary arterial remodelling by a process which is facilitated by reducing the expression of KCNK3 and loss of Kv1.5 function in pulmonary artery smooth muscle cells that contributes to dysregulation of K+ channels. This might be a hypoxia-independent event that can be observed in non-COPD smokers. Therefore, tobacco and hypoxia contribute to the development of PH by sharing similar intermediate mechanisms and could obviously enhance each other to accelerate the development of PH.

Our study is the first of its kind to comprehensively investigate the potential role of NH in the presence of chest CT-detected PAE in two well-characterised cohorts of smokers with or without COPD. However, some limitations to our study are worth mentioning. First, the population studied includes a selected sample of smokers attending a pulmonary clinic, albeit participant characteristics are similar to those reported previously in lung cancer screening trials [37,38] suggesting that the present study sample was representative of the wider smoking population. We cannot exclude a potential selection bias because we only studied subjects without other important causes of nocturnal hypoxemia such as heart failure, neuromuscular diseases, or the obesity-hypoventilation syndrome (OHS). On the other hand, we did not measure arterial blood gases and therefore we were able to inadvertently include some patients with OHS. Whether our findings are applicable to patients recruited from other clinical settings such as primary care remains to be determined. Second, we did not perform RHC, therefore we cannot estimate the concordance of the criteria used here to define PAE with the diagnosis of PH obtained by RHC. Third, instead of a full polysomnography (PSG), we have performed HSS and therefore we do not have the characteristics of the patients’ sleep. In addition, respiratory polygraphy underestimates the AHI score in severe OSA, and overestimates it in patients with very mild OSA. However, the purpose of this study was not to assess OSA or OVS prevalence, and we chose a low cut-off for apnoeic events (≥5 events·h^−1^) because with thought that this degree of sleep-disordered breathing would be relevant as a cause of nocturnal hypoxemia. Finally, as a cross-sectional study in nature, no conclusion can be established in our study regarding causality between NH and PAE.

## 5. Conclusions

This study shows an increased prevalence of PAE in smokers with or without OSA that varies according to the methodology used to define PAE. It is higher if the definition is based on the diameter of the PA using a sex-dependent measure and lower if the method is based on the PA:Ao ratio. Obesity, lower FEV_1_ and higher T90 are risk factors associated with PAE if the sex-specific PAD definition is used, while only T90 remains relevant if PAE is defined by the PA/Ao ratio. These results support the need to monitor nocturnal oximetry in smokers with PAE on CT imaging. Whether treatment of NH in COPD patients with PAE and confirmed PH has merit, is yet to be determined in future studies.

## Figures and Tables

**Figure 1 jcm-10-00489-f001:**
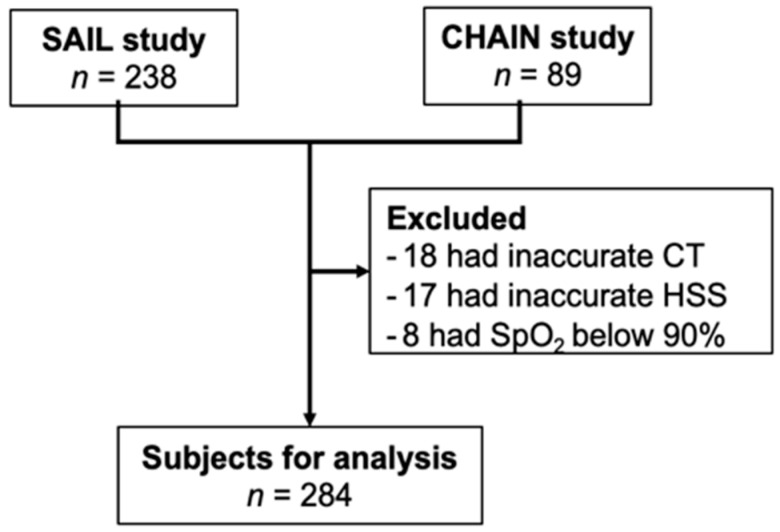
Flowchart of the study. SAILS = Sleep Apnoea in Lung Cancer Screening Study; CHAIN = Chronic Obstructive Pulmonary Disease (COPD) History Assessment in Spain study; CT = chest computed tomography; HSS = home sleep study.

**Figure 2 jcm-10-00489-f002:**
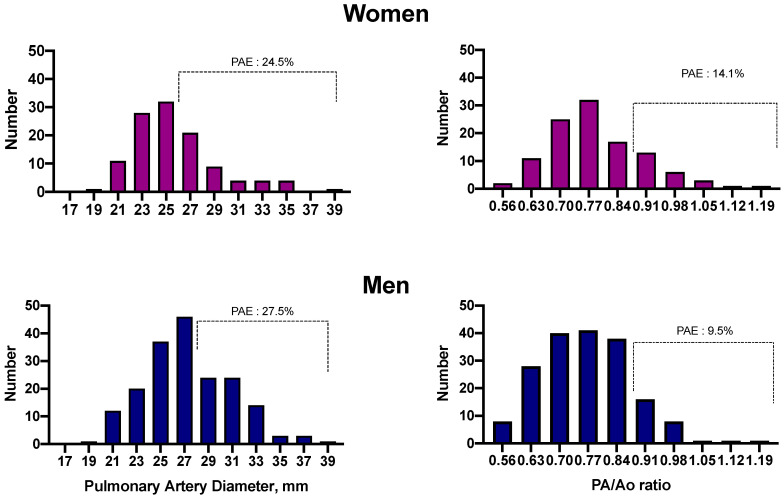
Distribution histogram for both women and men of the main pulmonary artery diameter (PAD) and ratio of PAD to ascending aorta diameter (PA/Ao ratio). Prevalent pulmonary artery enlargement (PAE) as defined by PAD and by PA/Ao ratio occurred in 24.1 and 15.4% of women and 29.1 and 9.3% of men, respectively.

**Figure 3 jcm-10-00489-f003:**
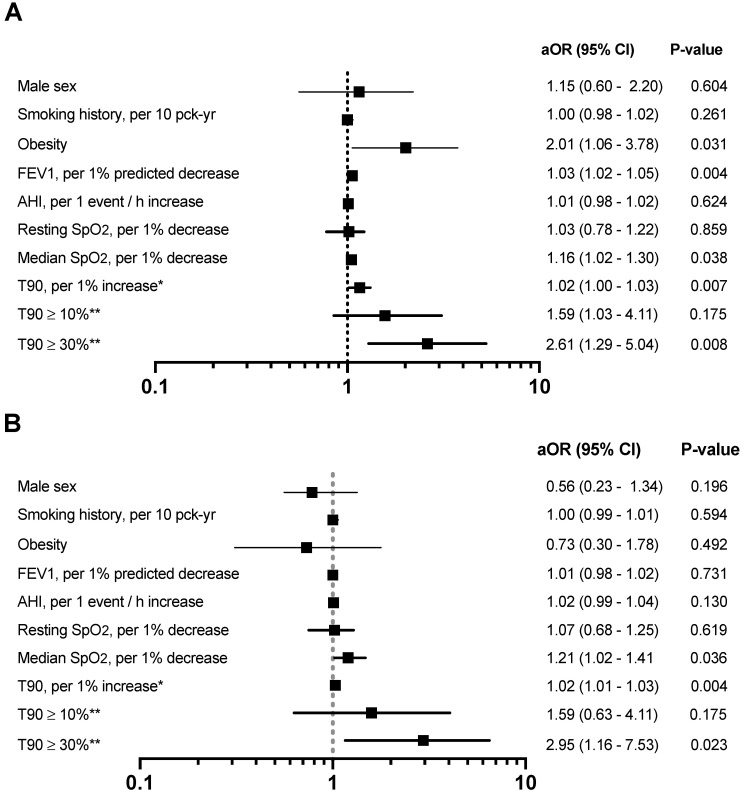
Forest plots of risk factors associated with pulmonary artery enlargement (PAE) as defined by the main pulmonary artery diameters (**A**) and as defined by PA:Ao ratio (**B**). Variables in the multivariate regression model included: age, gender, smoking status (current vs. ex-smoker), smoking history, obesity, FEV_1_% predicted, apnoea–hypopnoea index, resting O_2_ saturation, median O_2_ saturation and centre site. Abbreviations: aOR = adjusted odds ratio; CI = confidence interval, FEV_1_ = forced expiratory volume in one second, AHI = apnoea–hypopnoea index; T90 = percentage of sleep recording time with oxygen saturation < 90%. * OR estimates in the fully adjusted model instead of median O_2_ saturation. ** OR estimates in the fully adjusted model instead of median O_2_ saturation and T90.

**Table 1 jcm-10-00489-t001:** Baseline characteristics *.

Variable	All Subjects (*n* = 284)	No PAE (*n* = 209)	PAE (*n* = 75)	*p*-Value *
**Male**, *n*(%)	178 (63)	129 (62)	49 (65)	0.266
Age, years—mean (SD)	65 (7)	65 (7)	66 (7)	0.273
Smoking history Packs—year Current smokers, *n*(%)	42 (35–60) 142 (50)	40 (32–60) 105 (50)	50 (38–60) 37 (49)	0.027 0.893
Body mass index, kg/m^2^ Body mass index ≥ 30 kg/m^2^, *n*(%)	28 (25–31) 91 (32)	27 (25–31) 53 (26)	30 (25–34) 38 (51)	0.002 <0.001
Prevalent comorbidities, *n*(%) Hypertension Diabetes Hyperlipidaemia Coronary heart disease Atrial fibrillation	111 (39) 45 (16) 91 (32) 29 (10) 17 (6)	75 (36) 29 (14) 63 (30) 19 (9) 10 (5)	34 (46) 15 (20) 27 (36) 9 (12) 6 (8)	0.149 0.209 0.350 0.468 0.300
FEV_1_, % predicted -mean (SD) COPD, *n*(%)	84 (22) 171 (60)	88 (20) 124 (59)	74 (23) 47 (63)	<0.001 0.613
DL_CO_, % predicted—mean (SD)	83 (23)	84 (22)	81 (25)	0.310
Epworth sleepiness scale	6 (4–8)	6 (4–8)	6 (4–9)	0.069
Apnoea–hypopnoea index, events/h OSA, *n*(%)	10 (4–22) 167 (59)	10 (3–20) 124 (59)	11 (5–30) 43 (57)	0.273 0.763
Oxygen desaturation index, events/h	11 (4–22)	9 (4–19)	10 (5–28)	0.097
Resting O_2_ saturation, %	94 (92–95)	94 (92–95)	93 (92–94)	0.002
Median O_2_ saturation, %	92 (90–93)	92 (91–94	91 (89–93)	<0.001
T90, % T90 ≥ 10%, *n* (%) T90 ≥ 30%, *n* (%)	6 (1–30) 119 (42) 71 (25)	4 (1–23) 76 (36) 40 (19)	17 (1–69) 43 (57) 31 (41)	<0.001 0.002 <0.001
% of lung with emphysema	3 (1–7)	3 (1–7)	3 (1–6)	0.852
Pulmonary artery diameter, mm	26 (24–28)	25 (23–26)	31 (29–32)	<0.001
Aorta diameter, mm	34 (31–37)	33 (31–36)	36 (33–39)	<0.001
PA:Ao ratio	0.76 (0.68–0.84)	0.76 (0.66–0.81)	0.86 (0.79–0.92)	<0.001
PA:Ao > 0.9, *n*(%)	32 (11.6)	10 (4.8)	22 (29.0)	<0.001

Definition of abbreviations: PAE = pulmonary artery enlargement as defined by pulmonary artery diameter; Post-BD FEV_1_ = post-bronchodilator forced expiratory volume in the first second; COPD = chronic obstructive pulmonary disease; DLco = diffusing capacity of the lung for carbon monoxide; OSA = obstructive sleep apnoea as defined by an apnoea–hypopnoea index ≥ 5 events·h^−1^; T90 = percentage of sleep recording time with oxygen saturation < 90%; PA:Ao ratio = ratio of the diameter of the pulmonary artery to the diameter of the ascending aorta. Data presented as median (25th–75th percentiles), mean (SD), or number (%). * *p*-value differences were assessed using *t* tests or Mann–Whitney test for continuous variables and Pearson’s chi-squared tests for categorical variables.

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
