# Peer review of "Nocturnal Hypoxemia and CT Determined Pulmonary Artery Enlargement in Smokers"

_jcm, 2021, doi:10.3390/jcm10030489_

Round 1

Reviewer 1 Report

In this study the authors aim to test the hypothesis that in smokers with or without COPD, the presence of NH would be associated with PAE independently of the coexistence of OSA. Although the study is not novel and the given information already exists in the literature, the study is very well designed and the manuscript is well-written, giving the opportunity to increase evidence on pulmonary artery enlargement and factors associated with that in patients with respiratory diseases.

I have only one comment

The authors show that PEA can be found in a greater percentage of patients when it is defined by the main pulmonary artery diameters (27%) compared to the percentage when defined by PA:Ao ratio (11.6%). This could be probably related to the fact that in some patients Ao diameter could be also increased as a result of an aortic aneurism which is a marker of vascular disease. I believe that patients with aneurismatic enlargement of the Ao should be excluded since in these patients PAE could be also associated to cardiovascular parameters and cardiac function. On the contrary, when obvious cardiovascular abnormalities are absent PAE could be only be associated to respiratory parameters giving the real influence of hypoxemia and increase of pulmonary vascular resistance in PA. I do not as the authors to change the analysis, just to check their results of PA/Ao ratio excluding patients with abnormal Ao enlargement.

Author Response

We are very pleased you found our paper well designed and written. We have considered all your constructive comments and hope to have addressed them below.

We fully agree and thank you for this comment. You are right, the aorta diameter (Ao) could be increase as a result of an aortic aneurism. For this reason, we have excluded smokers with aortic aneurysm in the CT scan, a detail that it is included in the table of inclusion/exclusion criteria that can be consulted on page 2 of the supplementary material.

We like and thank your suggestion of re-analyzing our data excluding subjects with “abnormal Ao enlargement”.  There is not a consensus definition of “abnormal Ao enlargement” (AAE). However, in the most recent study about this topic, Marck et al (Mark DG, et al. Discriminatory Value of the Ascending Aorta Diameter in Suspected Acute Type A Aortic Dissection. Acad Emerg Med. 2019 Feb;26(2):217-225. doi: 10.1111/acem.13547), AAE was defined in adults with an aorta diameter > 41 mm. From the analysis of the raw data, we found that the median (IQR) of the diameter of the series as a whole is 34 (31-37) mm, with significant differences between smokers without and with PAE: 33 (31-36) vs 36 (33 -39), p <0.001. On the other hand, only 13 cases (4.5%) showed a diameter of Ao > 41 mm. Following your suggestions, we performed a sensitivity analysis excluding these 13 smokers, but the results were similar. Given that you also suggest not to change the whole analysis, if you wish, we now include the values of the aortic diameter in Table 1 (page 4/12), without including additional tables or figures that would unnecessarily extend the paper.

Reviewer 2 Report

The authors have analyzed ratio of the PAD to that of the aorta (PA:Ao ratio) from two prospective cohort studies that enrolled 284 smokers in lung cancer screening programs and com-pleting baseline home sleep studies and chest CT scans, and found the presence of nocturnal hypoxemia was associated with a higher prevalence of PAE on chest CT scan, independent of coex-isting OSA.

Overall, the manuscript is well written, and results and conclusions are clear and understandable. There are, however, several issues for the authors to address.

Criticisms

  1. It is apparent that COPD and OSA cause PH dependent of the disease severity. Severe COPD and or OSA could cause sever nocturnal hypoxemia and PH. Please show the patient profiles of severity of COPD according to the GOLD guidelines and analyze the PA:Ao ratio according to the disease severity. By the same token, the author should show the patient profiles of severity of OSA and analyze the association between the OSA severity and PA:Ao ratio.
  2. Interstitial lung disease is a disease associated with PH and nocturnal hypoxia (Respirology (2019) 24, 930–932). Similarly, combined pulmonary fibrosis and emphysema (CPFE), which shows normal limits or slightly reduced FVC and FEV1%, are also associated with PH. ILD and CPFD should be excluded from present study.
  3. Please discuss the pathogenetic mechanisms of smoking in developing of PH and/or nocturnal hypoxia independent of COPD and OSA.

Author Response

We are very pleased you found our paper well written and understandable. We have considered all your constructive comments and hope to have addressed them below.

Criticism 1. 

We agree that severe COPD and or OSA could cause severe nocturnal hypoxemia and PH. Interesting in that our population included smokers who had come to the clinic primarily to screen for lung cancer, hence the degree of airflow obstruction is globally mild (FEV1 84% predicted).

Following the reviewer's recommendations, we now indicate in the new revised version in Table S4, the distribution of COPD and OSA patients according with GOLD classification and AHI index respectively.  We have explored the association of between COPD and OSA severity and pulmonary enlargement as defined by sex-specific pulmonary artery diameter (PAD) or PA:Ao.  As expected by the results shown in figure 3, severe COPD had a higher PAE as assessed by PAD prevalence than mild-moderate COPD but there was no differences when PAE was defined by PA:Ao. We include a comment of this results also in the text of the new version (page 6, first paragraph).

Criticism 2. 

Good point. No participant included in this analysis had previously been diagnosed with ILD or showed significant interstitial involvement on CT scan. We include this exclusion criterion in the corresponding table of the supplementary material.

Criticism 3. 

The reviewer has taken careful note that among smokers without COPD or OSA, 26% and 11% have PAE defined by the diameter of the pulmonary artery or by the PA: Ao ratio respectively (page, point 3.2).

From the results of our multivariable analysis, it is concluded that the accumulated number of pack-years or the condition of active smoker is not associated with incident PAE. We can therefore only speculate that there must be a genetically based response to the effect of tobacco smoke on the pulmonary vasculature. It is well known that hypoxia drives pulmonary hypertension (PH) by promoting pulmonary arterial remodeling and dysregulation of voltage-gated K+ channels. Interestingly, Sevilla-Monter et al (Am J Respir Crit Care Med 2020 Dec 11), showed that cigarette smoking contributes to pulmonary arterial remodeling and induction of arterial smooth muscle and adventitial fibroblast senescence. This process is facilitated by the under-expression of KCNK3 and loss of Kv1.5 function in pulmonary artery smooth muscle cells that contributes to dysregulation of K+ channels and increase cytosolic Ca++ concentration in smooth muscle. This might be a hypoxia-independent event that can be observed in non-COPD smokers. Therefore, tobacco and hypoxia contribute to the development of PH by sharing similar intermediate mechanisms and could obviously enhance each other to accelerate the development of PH.

As suggested, we now include a comment about this topic in the discussion section (page 9/12) and included 2 additional references.

Round 2

Reviewer 1 Report

I have no additional comments

Reviewer 2 Report

The authors corrected the comments appropriately.